# The development of a safe opioid use agreement for surgical care using a modified Delphi method

**Cassandra B. Iroz**[1], **Willemijn L. A. Schäfer**[1], **Julie K. Johnson**[1], **Meagan S. Ager**[2], **Reiping Huang**[1], **Salva N. Balbale**[1,3,4], **Jonah J. Stulberg**[5]*, on behalf of the Opioid Agreement Delphi Group[¶]

**1** Northwestern Quality Improvement, Research, & Education in Surgery (NQUIRES), Department of Surgery, Northwestern University Feinberg School of Medicine, Chicago, IL, United States of America, **2** Mathematica Policy Research, Chicago, IL, United States of America, **3** Division of Gastroenterology and Hepatology, Department of Medicine, Northwestern University, Chicago, IL, United States of America, **4** Center of Innovation for Complex Chronic Healthcare, Health Services Research & Development, Edward Hines, Jr. VA Hospital, Hines, IL, United States of America, **5** Department of Surgery, University of Texas McGovern Medical School, Houston, TX, United States of America

¶ Membership of the Opioid Agreement Delphi Group is provided in Acknowledgments.
* jonah.j.stulberg@uth.tmc.edu

**Data Availability Statement:** Our quantitative data are publicly available at https://doi.org/10.21985/

## Abstract

### Background

Opioids prescribed to treat postsurgical pain have contributed to the ongoing opioid epidemic. While opioid prescribing practices have improved, most patients do not use all their pills and do not safely dispose of leftovers, which creates a risk for unsafe use and diversion. We aimed to generate consensus on the content of a "safe opioid use agreement" for the perioperative settings to improve patients' safe use, storage, and disposal of opioids.

### Methods

We conducted a modified three-round Delphi study with clinicians across surgical specialties, quality improvement (QI) experts, and patients. In Round 1, participants completed a survey rating the importance and comprehensibility of 10 items on a 5-point Likert scale and provided comments. In Round 2, a sub-sample of participants attended a focus group to discuss items with the lowest agreement. In Round 3, the survey was repeated with the updated items. Quantitative values from the Likert scale and qualitative responses were summarized.

### Results

Thirty-six experts (26 clinicians, seven patients/patient advocates, and three QI experts) participated in the study. In Round 1, >75% of respondents rated at least four out of five on the importance of nine items and on the comprehensibility of six items. In Round 2, participants provided feedback on the comprehensibility, formatting, importance, and purpose of the agreement, including a desire for more specificity and patient education. In Round 3, >75% of respondents rated at least four out of five for comprehensibility and importance of

n2-12je-4d95 and our qualitative data are publicly available at https://doi.org/10.21985/n2-4hza-ec42.

**Funding:** The research presented here was supported by a grant from the Agency for Healthcare Research and Quality (grant number R18HS027331). Jonah Stulberg (JJS) is the principle investigator on the grant and all other authors were supported by the grant. The funders had no role in study design, data collection and analysis, decision to publish, or preparation of the manuscript. Funder website: https://www.ahrq.gov/.

**Competing interests:** The authors have declared that no competing interests exist.

all 10 updated item. The final agreement included seven items on safe use, two items on safe storage, and one item on safe disposal.

## Conclusion

The expert panel reached consensus on the importance and comprehensibility of the content for an opioid use agreement and identified additional patient education needs. The agreement should be used as a tool to supplement rather than replace existing, tailored education.

## Introduction

Postoperative prescription opioid use has contributed to the ongoing opioid epidemic. A substantial portion of opioid prescriptions in the United States are for acute pain management after surgery [1, 2], and can still lead to chronic use [3–5]. On average, 70% to 90% of opioid pills prescribed following surgery remain unused after the initial pain episode [6–11]. Over 70% of surgery patients do not dispose of their unused opioids [9, 12], increasing the risk for non-medical use [13–15], adverse drug events, and use of other illicit drugs [13, 16]. Therefore, safe disposal is key to preventing adverse events and diversion [17–20].

Engaging patients and clinicians is critical to address the opioid epidemic. While various initiatives have successfully reduced opioid prescribing [21–25], there is an ongoing need for clinicians to better engage patients in safe opioid use practices [26]. Opioid use agreements, also known as opioid contracts [27], are one tool to engage patients, are well-established in primary care [28–32], and have been shown to increase patients' participation in their care [33] and reduce opioid use [28, 31]. To date, studies on these agreements in pain clinics and primary care have shown reductions of 7% to 23% in opioid misuse [34–37], although specific effects on disposal have not been measured [38]. However, currently available opioid use agreements have shown some shortcomings in their design. A national survey of providers showed that most agreements were written above recommended reading levels, highlighting the need for a more patient-centric approach [39].

Use of agreements in acute pain management offers a logical extension of current practices from chronic pain management. However, agreements have not been studied in the perioperative setting. Opioid use agreements have primarily been used in populations with long-term opioid use to treat chronic pain, whereas opioids prescribed following surgery are intended to treat acute short-term pain [27]. Additionally, the patient-clinician relationship can differ between primary care where there are often long-term relationships versus surgery, which is characterized by more episodic, acute care. It is therefore important that we understand what is needed from patient, provider, and quality improvement (QI) perspectives in a surgery-specific context.

We therefore aimed to develop and generate consensus on the content of a safe opioid use agreement to improve safe use, storage, and disposal of opioids prescribed after surgery.

## Methods

### Study design

We conducted a modified three-round Delphi study to generate consensus on the content of a safe opioid use agreement among a diverse stakeholder panel including patients, surgeons, nurses, pharmacists, and QI experts. The Delphi method is a reliable way to gather consensus

from a group of experts [40]. It is a flexible approach, and while there are no universal guidelines, there are recommendations for best practices [41, 42]. In a Delphi study, experts rate items then reevaluate their ratings until consensus is reached. The process for our study took place from July to October 2020.

## Sample

We selected experts for our panel using purposeful and snowballing sampling techniques, ensuring representation of each stakeholder group, and including participants with relevant clinical expertise. This was done by reaching out to our research team's professional network and targeting individuals who were interested in, or had participated in, opioid reduction initiatives. We invited surgeons and inpatient and outpatient nurses from a variety of specialties (orthopedics, urology, trauma, gynecology, surgical oncology, and general surgery), pharmacists, QI experts, and patients, from one single healthcare system. We also asked participants to suggest additional stakeholders to include, particularly within the nursing groups. All clinician participants worked with adult patients at a large, private, urban academic medical center in the United States. Patient representatives were members of our healthcare system's Patient Family Advisory Council and were asked to participate and provide their perspective on how a patient might interpret and perceive the agreement.

## Rounds of the Delphi study

We followed a three-round modified Delphi approach (Fig 1). First, our research team developed a draft agreement including ten items based on existing opioid use agreements used in primary care [43–47]. Items included in the draft agreement were chosen through discussions within the research team and were based on publicly available opioid agreements and our previous research on surgical opioid reduction within our healthcare system. For example, our previous work showed that communicating and setting expectations about pain relief as well as discussing safe disposal of leftover opioids were important areas for improvement, so they were included in the draft agreement [48, 49]. We developed a survey to rate the importance and comprehensibility of each item of the draft agreement. The survey included free text fields for participants to add explanations on the importance and/or comprehensibility. The survey also asked if there were any topics that were not covered in the draft agreement that they believed should be included.

**Pilot testing.** Pilot testing is recommended before starting a Delphi study [41, 50]. We pilot tested the survey with individuals of various ages and with various expertise in our personal networks (n = 7) and made changes based on their feedback to the survey comprehensibility, the agreement content, and its length. Based on respondent feedback, we revised response categories to reflect validated five-point Likert scales ranking comprehensibility as "Very poor", "Poor", "Fair", "Good", or "Very good", and importance as "Not important", "Slightly important", "Moderately important", "Important", or "Very important". We also added a brief introduction to the agreement, reworded several items, and changed the order.

**Round 1.** In Round 1, the expert panel received the survey (S1 File) via email and completed it in Qualtrics software (Qualtrics, Provo, UT). The survey was open for 1.5 months and participants were sent one reminder. We then distributed feedback reports to the participants, including a summary of the survey results compared to their individual answers so they could reflect on their responses and revise their opinions for the following rounds (an example feedback report is available in S2 File).

**Round 2.** In Round 2, we invited 15 members of the panel to participate in a focus group. Focus group participants were selected to ensure representation of the different stakeholder

**Pilot**
- **Methods**: Pilot survey
- **Participants**: 7 pilot reviewers

**Round 1**
- **Methods**: Survey
  - 5-point Likert Scale "Very poor", "Poor", "Fair", "Good", or "Very good" for comprehensibility
  - 5-point Likert Scale "Not important", "Slightly important", "Moderately important", "Important", or "Very important".
  - Free text for explanations
  - Asked about additional items to include
- **Participants**: 36 experts

**Round 2**
- **Methods**: Focus Group
  - Discussion of items with less than 85% consensus
- **Participants**: 15 experts

**Review**
- **Methods**: Review by Patient Education Specialist
- **Participants**: One independent Patient Education Specialist

**Round 3**
- **Methods**: Survey
  - 5-point Likert Scale "Very poor", "Poor", "Fair", "Goood", or "Very good" for comprehensibilty
  - 5-point Likert Scale "Not important", "Slightly importnat", "Moderatley important", "Important", or "Very important"
  - Free text for explanations
  - Additional items to include
- **Participants**: 29 experts

**Fig 1. Outline of the Delphi study.** The modified Delphi study occurred in three rounds, starting with a survey, followed by a focus group, and ending with a final survey.

groups and surgical specialties. The group was limited to 15 participants to allow for interactive conversations. The focus group lasted 90 minutes, was conducted virtually, and was audio recorded and transcribed for analysis. Participants had the opportunity to provide input through the chat function. There were four moderators who were members of the research team and ensured 1) discussion of comments typed in the chat function, 2) an opportunity for all participants and participant groups to talk and state their opinions, and 3) strict time management. The focus group was semi-structured, and the presentation slides served as the moderator guide. The moderators presented background information on surgical opioid prescribing rates and research on the importance of patient education. The participants were then shown the quantitative and qualitative responses from the Round 1 survey and asked to discuss all items where less than 85% of respondents rated at least four out of five on the Likert scale (i.e., "Good" or "Very Good" for comprehensibility and "Important" or "Very Important"). While greater than 75% was defined as acceptable a priori, for the focus group we chose to discuss items with less than 85% agreement because agreement was high on the Round 1 survey, and we wanted to receive feedback on how to improve the lower scoring items. The moderators then asked the participants open-ended questions to elicit their opinions on the importance and/or comprehensibility of each of the selected items. The moderators specifically called out participants from all stakeholder groups to ensure that all groups were represented.

We also used the polling function to measure agreement during the focus group, and results were summarized along with the qualitative comments and shared with participants before the Round 3 survey. The focus group concluded with a discussion on the purpose of the agreement, length, introductory text, and additional topics to cover in the agreement.

Following the focus group, the research team used the transcript to update the agreement. This included creating a document summarizing the areas where additional patient education was needed to support effectiveness of the agreement (S3 File). The language of the agreement was then reviewed and edited by a Patient Education Specialist, who was independent from the research team, to reflect a sixth to eighth grade reading level.

**Round 3.**   In Round 3, we repeated the survey from Round 1 with the revised items. The survey was sent to the entire 36-member panel, along with a feedback report that included detailed explanations of changes made. The survey was open for one month and participants were sent up to one reminder. In the final agreement, we included items for which more than 75% of participants rated at least four out of five on the Likert scale for importance and comprehensibility. There is no standard definition of consensus for Delphi studies with recommendations ranging from 51% to 100%, but 75% is a common benchmark [41, 42, 51].

## Data analysis

Survey responses were exported from Qualtrics (Qualtrics, Provo, UT) to Microsoft Excel (Microsoft Corporation, Version 2304) and quantitative values were summarized using descriptive statistics. Qualitative responses from the free text fields in the Round 1 and Round 3 surveys as well as the transcript of the focus group from Round 2 were summarized using a simple thematic analysis approach and lean coding [52, 53]. First, two researchers (CBI & WLS) reviewed all qualitative comments provided on the Round 1 and Round 3 surveys and the transcript from the focus group. Second, the two researchers discussed the qualitative data to develop codes inductively which were applied to the data through group discussion. For example, quotes relating to desired reading level of the items were labeled as "reading level." Third, they created a table grouping the broad codes to identify overarching themes [52]. For example, the "reading level" code was assigned to the theme "comprehensibility." Finally, themes and example quotes were discussed with two additional team members and refined for clarity (JKJ & SNB).

## Ethics

Northwestern University's Institutional Review Board reviewed the study and determined that it did not qualify as human subjects research (STU00212619). All participants were informed about the purpose of the research, procedures, potential risks and benefits, and that participation was fully voluntary and could be stopped at any time. Each expressed their consent to participate in writing for the survey and again verbally for the focus group.

## Results

Our panel consisted of 36 experts including nine pharmacists, seven nurses, seven surgeons, seven patients/patient advocates, three QI experts, one nurse practitioner, one anesthesiologist, and one emergency medicine physician. Thirteen experts participated in all three rounds (Table 1).

## Round 1 results

In the Round 1 survey, more than 75% of respondents rated nine items as "Important" or "Very Important" and six items as "Good" or "Very good" for comprehensibility (Table 2). We

**Table 1. Participants for each round of the Delphi study.**

| Role | Round 1 | Round 2 | Round 3 |
|---|---|---|---|
| Pharmacists | 9 | 2 | 6 |
| Nurses | 7 | 2 | 7 |
| Surgeons | 7 | 4 | 7 |
| Patient/Patient Advocates | 7 | 3 | 5 |
| QI Experts | 3 | 2 | 3 |
| Nurse Practitioner | 1 | 1 | 0 |
| Anesthesiologist | 1 | 1 | 0 |
| Emergency Medicine Physician | 1 | 0 | 1 |
| **Total Responded** | **36** | **15** | **29** |
| Total Invited | 36 | 15 | 36 |
| **Response Rate** | **100%** | **100%** | **80.6%** |

summarized the qualitative comments into ten themes (Table 3). Qualitative feedback from participants revealed a desire for greater specificity, issues with comprehensibility including concerns about medical terminology such as "respiratory depression", concerns about the effectiveness of the agreement in practice, complexity of wording, questions on the purpose of the agreement, concerns about Item 10 related the Prescription Monitoring Program (PMP), and a desire for language on shared responsibility. Minor updates were made in the agreement (i.e., correcting spelling and grammar) prior to the Round 2 focus group.

A total of 19 participants (53%) responded "Yes" to the question asking if there were additional topics that were not covered that they believe should be included. Participants reported the need for additional patient education, including information on alternative pain management strategies, information about how to safely discontinue opioid use, and additional risks of opioids including sedation, interaction with alcohol, and the risk for addiction. Further, participants wanted more specificity on how patients should communicate with their healthcare provider, and the types of information that patients should share, such as notifying their provider about over-the-counter medications. Another suggestion was to include an item about patients not receiving opioid prescriptions from other providers. Finally, one participant was concerned that the agreement would make patients hesitant to take their prescribed opioids.

## Round 2 results

Fifteen members of the expert panel were invited to and participated in the focus group (Table 1). Themes from the focus group are summarized in Table 3. During the focus group, the panel discussed how to improve the comprehensibility for Items 2, 3, and 9. For Item 5 on opioid interactions, the group discussed the wording and layout as well as which medications were important to include in the list with potential interactions. In both Rounds 1 and 2, some participants suggested adding more detail to various items. Through discussion in the focus group, it was decided that by keeping the items more general, the agreement could be more easily adapted to different surgical specialties and practices. The participants identified additional education needs for Items 2, 3, 5, 7, and 9. Examples of such education included when, how, and with whom to communicate about pain, individualized examples of potential drug interactions, and information about how to safely store and dispose of opioids. The focus group also included discussion on the potential effectiveness of the agreement, purpose of the agreement, and questions about the Prescription Monitoring Program.

**Table 2. Percentage of participants who rated at least four out of five for Round 1 and Round 2.** Green cells represent items that met the threshold for consensus (>75% of respondents rating at least four out of five) and red cells represent items that did not meet the threshold for consensus.

| Item as presented for Round 1 | Round 1 survey results | | Item as presented for Round 3 | Round 3 survey results | |
|---|---|---|---|---|---|
| | Considered important | Considered comprehensible | | Considered important | Considered comprehensible |
| 1. I understand that an opioid is a medication to treat pain. | 100% | 92% | 1. I understand that an opioid is a medicine used to treat pain. | 86% | 93% |
| 2. I will communicate fully with my doctor about the intensity of my pain, the effect of the pain on my daily life, and how well the medicine is helping to relieve the pain | 86% | 78% | 2. I will tell my doctor about my pain. | 83% | 79% |
| 3. I agree that I will use my medicine as prescribed. If I use my medicine at a greater rate it could lead to drug overdose causing severe sedation and respiratory depression and death. | 100% | 67% | 3. I understand that if I use more of my opioid pain medicine than prescribed it could cause an overdose and death. | 100% | 90% |
| 4. I will inform my doctor of all medications I am taking, including any herbal/health supplements. | 86% | 92% | 4. I will tell my doctor all of the medicines I am taking, including any herbal/health supplements. | 93% | 90% |
| 5. I understand that there can be serious side effects when I use the opioid medications when I am taking other medications, such as Valium or Ativan; other opioid medicines; sedatives such as Soma, Xanax, Fiorinal; antihistamines like Benadryl; herbs, alcohol, and cough syrup | 89% | 64% | 5. I understand that there can be serious side effects if I use my opioid pain medicine while I am using other medicines or substances, such as:<br>□ Other opioid pain medicines (e.g. oxycodone, codeine)<br>□ Benzodiazepine sedatives (e.g. diazepam (Valium®), lorazepam (Ativan®), alprazolam (Xanax®))<br>□ Muscle relaxants (e.g. carisoprodol (Soma®), cyclobenzaprine (Flexeril®))<br>□ Headache pain medicine containing a butalbital (e.g. Fiorinal®)<br>□ Antihistamines (e.g. diphenhydramine (Benadryl®))<br>□ Cough suppressants<br>□ Alcohol<br>□ Herbal supplements | 90% | 79% |
| 6. I will safeguard my pain medication from loss, theft, or unintentional use by others. | 94% | 86% | 6. I will keep my opioid pain medicine safely stored to avoid loss, theft or use by others. | 90% | 93% |
| 7. I understand that lost or stolen medications will not be replaced. | 89% | 89% | 7. I understand that lost or stolen opioid pain medicine may not be replaced. | 79% | 86% |
| 8. I understand that the pain medication is strictly for my own use. I will never share my medication with anyone because it may endanger that person's health and is against the law | 94% | 97% | 8. I understand that the opioid pain medicine is strictly for my own use. I will never share my opioid medicine with anyone because it may harm that person's health, and it is against the law. | 97% | 93% |
| 9. I will dispose of unused opioid medicines as recommended by my doctor or pharmacy when I am done using them to treat my pain from surgery. | 89% | 75% | 9. I will safely dispose of the unused opioid medicine when I am done using it to treat my pain from this surgery. | 90% | 79% |
| 10. I understand that my doctor is required by law to check the state database, which lists other opioid prescriptions that I receive before writing a new prescription for an opioid medicine. | 72% | 75% | 10. I understand that my doctor is required by law to check the state records for other opioid prescriptions that I receive before writing a new opioid pain medicine prescription for me. | 83% | 86% |

**Table 3. Summary of qualitative feedback.** Themes are presented with example quotations from each round.

| Theme | Round 1 | Round 2 | Round 3 |
|---|---|---|---|
| Need for more specificity or more detail | *"What does this include? Patients can drink alcohol or tea, or they can 'vape' or smoke. What's included?"- Patient* | *"Do you want to put a parameter on when they're telling their doctor about pain? Because I can't imagine that, let's say a surgeon wants to hear from every single patient every day if they're not experiencing pain"-Pharmacist* | *"No. 2 is too open ended. Type of pain? Location of pain? Severity of pain?"- Patient* |
| Comprehensibility • Generic comments about readability | *"This statement is more clear and concise."- Surgeon* | *"How about something more simple [for Statement 2]. . .'I will contact my doctor's team if my pain is too much.'"- Surgeon* | *"Fantastic job making these concise and easy to understand! I think this will work very well and should be easy to understand for everybody."- Patient* |
| • Technical language/ Differentiating opioids from other medications/ Generic vs. brand name | *"Throughout, 'pain medication' and 'opioid' seem to be used interchangeably. Some patients will not know what an 'opioid' is, so they may not understand the statement."- Patient* | *"I think you're making some assumptions about what patients understand. Do patients understand. . .what an opioid is, what overdose is, and why overdose should be avoided, what's the patient likely to experience if they experience overdose?"- Patient* | *"Do patients understand that 'opioid' and the prescribed Norco are essentially the same thing? Should you start with 'opioids are a class of medications used to treat pain. I am prescribing, X which is within the class of pain medications' or something like this?"- QI expert* |
| • Reading level/health literacy | *"I think that the comprehensibility could be improved by bringing this to the sixth-grade reading level."- Anesthesiologist* | *"We worked with the Academy at [Hospital System] just to make sure that everything was at fifth grade reading level. So I know that sometimes that they can be helpful because they have a specialist that works with this"- QI expert* | |
| Importance and effectiveness of the agreement | *"All the statements are good and should be included. Most compliant patients will follow your instructions. Those patients that are not will say they will comply then do what they want."- Patient* | *"The goal is to change behavior. If they break this that's identifiable to a physician. It seems like you want to focus on the behavior as opposed to what the doctors are doing."- Patient* | *"I think this current draft captures the items that are of significant importance in getting patients to buy into the tenets of safe use of opioids in the pain management regimen. I think we need to be aware that there may be some measure of disconnect as a number of patients may assign a diminished level of importance relative to that which we apply. . ."- Pharmacist* |
| Length of the agreement/too wordy/too long | *"Wording too complex. Not simple understandable language, short sentences, etc."- QI expert* | | |
| Formatting | | *"I understand that other medications can cause serious side effects when taken with opioid medications, including the following. And then to put them in bullets"- Surgeon* | *"Number 8 could potentially be split into two separate items?"- Nurse* |
| Purpose of the agreement | *"As a patient, this seems like a 'CYA' statement to me. . .I assume the legal aspect is covered anyways and this is more about providing actionable guidance for people."- Patient* | *"I think we're hovering on this question of what's the boundary between education and an agreement that signifies I've received education and I comprehended it"- Surgeon* | |
| Connotation of certain words | *"Does the use of the term 'honestly' presume dishonesty prior to this agreement?"- Surgeon* | *"Death is the flashpoint term. That's the term that you can't look away from."- Pharmacist* | |
| Patients do not know and do no need to know about the PMP | *"I feel some patients may not be privy to this law existing."- Pharmacist* | *"Why is it important for patients to know that a doctor is doing that? The doctor will check the database and make a decision based on what he finds and communicate that decision with the patient. Why does the patient need to know how he came to that information? Or that he is required to do that?"- Patient* | |

(*Continued*)

**Table 3.** (Continued)

| Theme | Round 1 | Round 2 | Round 3 |
|---|---|---|---|
| Concerns about alignment with current practice | *"Statement 10 may be problematic for physicians/advanced practice providers. Perhaps re-word the statement to 'I understand that my doctor may be required to check the state database.'"- Surgeon* | | |
| Shared responsibility | *"I would like to see the agreement use more shared responsibility language."*- Surgeon | | |

The Patient Education Specialist reviewed the agreement after the focus group and made some wording adjustment (such as changing "medication" to "medicine) and suggested we include generic as well as brand names for medications listed in Item 5.

## Round 3 results

For the Round 3 survey, 29 of the original 36 (80.6%) experts responded (Table 1). All ten items met the final threshold for inclusion with greater than 75% of respondents rating the item as "Important" or "Very important" and "Good" or "Very good" for comprehensibility (Table 2). There were far fewer qualitative comments on the Round 3 than the Round 1 survey. The remaining comments contained a continued desire for specificity, comments on the improved comprehensibility with additional slight modifications, a minor formatting suggestion, and comments on the purpose of the agreement (Table 3). Minor edits were made based on these comments and the final safe opioid use agreement is available in Fig 2. The final agreement included seven items on safe use, two items on safe storage, and one item on safe disposal.

## Discussion

Through a modified Delphi study, a diverse expert panel of clinicians, QI experts, and patients reached consensus on the content of a safe opioid use agreement for the perioperative setting. Ten items on the safe use, storage, and disposal of opioids prescribed for postoperative pain management were included and refined throughout the three rounds.

Our modified Delphi study was conducted in three rounds, with a focus group in the second round to engage our expert panel in a live exchange of ideas and gather opinions from different stakeholder groups. While the traditional Delphi approach relies on participants never interacting with one another [54, 55], including a focus group is a common modification, especially in healthcare quality studies [56]. One of the benefits of the Delphi approach is that the "grassroots involvement" can be highly motivating for participants [54].

In a few instances, the percentage of participants who rated at least four out of five for comprehensibility or importance decreased from Round 1 to Round 3 (Table 2). It is important to note that the sample size was larger (n = 36) in Round 1 compared to Round 3 (n = 29). Despite these decreases, there were still greater than 75% of participants rating at least four out of five, which met our a priori definition of consensus. We also found no additional comments in the qualitative feedback from Round 3 that would indicate any new concerns.

A common theme in the qualitative comments was a desire for more specificity or more detail on the included items. Perhaps the most important lesson learned from the Delphi study was the need for the research team to be clear that the purpose of the agreement was not to replace existing patient education on safe use, storage, and disposal of prescription opioids, but to act as a behavioral modification tool to improve compliance with safe opioid practice

## Safe Opioid Use Agreement

Your doctors and nurses want to help you manage your pain. We ask all patients who have had surgery and received an opioid prescription to sign this "Safe Opioid Treatment Agreement". Opioid pain medicines can work very well to relieve your pain, but they have risks to you and those around you. This agreement helps you and your healthcare team to use your opioid medicine safely.

For my new opioid pain medicine prescribed for this surgery, I agree to the following:

☐ I understand that an opioid is a medicine used to treat pain.

**Using this prescribed opioid pain medicine**

☐ I will tell my doctor about my pain.

☐ I understand that if I use more of my opioid pain medicine than prescribed it could cause an overdose and death.

☐ I will tell my doctor all of the medicines I am taking, including any herbal/health supplements.

☐ I understand that there can be serious side effects if I use my opioid pain medicine while I am using other medicines or substances, such as:
- Other opioid pain medicines (e.g. oxycodone, codeine)
- Benzodiazepine sedatives (e.g. diazepam (Valium®), lorazepam (Ativan®), alprazolam (Xanax®))
- Muscle relaxants (e.g. carisoprodol (Soma®), cyclobenzaprine (Flexeril®))
- Headache pain medicine containing a butalbital (e.g. Fiorinal®)
- Antihistamines (e.g. diphenhydramine (Benadryl®))
- Cough suppressants
- Alcohol
- Herbal supplements

☐ I will keep my opioid pain medicine safely stored to avoid loss, theft or use by others.

☐ I understand that lost or stolen opioid pain medicine may not be replaced.

☐ I understand that the opioid pain medicine is strictly for my own use. I will **never** share my opioid medicine with anyone because it may harm that person's health, and it is **against the law.**

**Disposing of any of this unused opioid pain medicine**

☐ I will safely dispose of the unused opioid medicine when I am done using it to treat my pain from this surgery.

**What my doctor will do**

☐ I understand that my doctor is required by law to check the state records for other opioid prescriptions that I receive before writing a new opioid pain medicine prescription for me.

**Fig 2. The safe opioid use agreement.** The Safe Opioid Use Agreement for the perioperative setting, as developed by the Delphi study participants included ten statements and an introduction describing the purpose of the agreement.

and to supplement thorough pain management education. Despite this consensus during the focus group, comments from the survey in Round 3 indicated that participants might not have understood this goal and wanted to continue to add details to the agreement to ensure all the information patients need was included in the document. This clarity on the purpose of the agreement needs to be a central message during the implementation of the agreement. There is a risk that healthcare providers will forego personalized patient education during the clinical

visit or remove existing patient education tools because of the presence of the agreement, which would be antithetical to the intended use.

Due to the recurring theme of a desire for more specificity or more detail, we outlined the additional patient education needs suggested by the expert panel (S3 File). The purpose of this document is to cue the person administering the agreement to additional education they might need to provide to ensure the patient is able to understand and comply with the item in the agreement. Clinical practice guidelines recommend "patient and family-centered, individually tailored education" [57]. Clinicians will continue to need to have individualized, patient-centered discussions on opioid safety and this format can help guide that discussion. In the next phase of implementation, we will interview clinicians about this document and other needs for opioid patient education materials.

Lack of disposal of unused prescription opioids is a significant problem and creates risk for diversion and misuse. Despite efforts to reduce opioid prescribing [21–25] there remains a need to focus on patient engagement in safe opioid use. The number of pills used for a given pain episode can vary greatly [9], meaning there will likely always be a risk of unused pills, potential for diversion, and need for safe disposal. Item 9, which was specifically about disposal was rated as "Important" or "Very Important" by 90% of the participants, underscoring the necessity to discuss safe disposal with patients. Additionally, Item 8, which relates to the issue of diversion, was rated "Important" or "Very Important" by 97% of the participants.

Reducing opioid use in surgery requires multicomponent efforts to be successful [24]. The safe opioid use agreement, developed through the Delphi study, can serve as one piece of that multifaceted strategy. As we found in this study, the agreement cannot stand alone, but should be used as one component of a broader strategy to improve prescribing and patient education on pain management strategies. Additionally, the agreement might be especially helpful for patients who are at higher risk of opioid misuse, including those who use opioids before surgery [58].

Other studies have found limited effectiveness of opioid agreements in primary and chronic care settings [29, 38, 59]. Limitations of these previous studies include inconsistent use of the agreement, sampling bias, lack of a consistent definition of opioid misuse or abuse, and few studies provided sufficient description or shared the actual text of their agreement. Approaches to reducing opioid addiction have been categorized as primary (preventing new cases of opioid addiction), secondary (identifying early cases of opioid addiction), and tertiary (ensuring access to effective addiction treatment) [60]. Previous studies have mostly focused on secondary prevention, working with populations who chronically use opioids, whereas this study was intended to develop an agreement for primary prevention, reducing the risks for patients prescribed opioids for acute pain management. To increase the likelihood our agreement would be effective, we followed a robust Delphi approach to develop content that was relevant to the patient population and easy to understand. We engaged a diverse panel of key stakeholders from various professions and specialties to generate consensus on the content of the agreement and had the language reviewed by a Patient Education Specialist. The effectiveness of our agreement is still unknown. The next step for our research team is to study the effectiveness of the agreement at improving safe use, storage, and disposal of opioids prescribed for postoperative pain management.

## Strengths and limitations

We noted a few limitations of our study. First, the agreement was developed only in English, limiting its use in diverse patient populations. Second, we only sought experts from one healthcare system, which perhaps limits the applicability in other settings. This focus on our own

healthcare system was purposeful, as we wanted to develop, implement, and test the agreement locally before disseminating to other settings. Reliability and validity of results from Delphi studies have been questioned, but including experts in the field of study strengthens the validity [41, 50, 61]. It is possible, and perhaps likely, that a similar experts panel with different participants would have developed a slightly different final agreement. Third, not all 36 members of the expert panel participated in the focus group. This was intentional to allow for more interactive discussion. However, one limitation with this approach is that the rest of the panel did not hear what was discussed in the focus group and responses in Round 3 continued to request more detail, a topic discuss in the focus group. Fourth, while we provided feedback reports, we cannot guarantee that participants reviewed them and considered their peers' feedback in their Round 3 responses. Finally, the nature of hierarchical relationships in medicine between the various stakeholder groups might mean that some individuals did not speak freely in the focus group (e.g., patients might have resisted disagreeing with physicians). However, given the purpose of the study (i.e., consensus building) it was important to have all stakeholder groups in one focus group and we believe we mitigated the bias through the moderation process.

There are some notable strengths in our approach. First, we involved a diverse group of stakeholders with different perspectives, expertise, and roles. The inclusion of pharmacists, nurses, surgeons, QI experts, and physicians in other specialties (i.e., anesthesiology and emergency medicine) provided a well-rounded clinical perspective. Including patients and patient advocates was also essential to understand the impact this agreement might have on patients and how they understand the agreement. Second, we pilot tested our survey which is recommended as a best practice [41, 50]. Third, we had a high response rate throughout the entire study, particularly since attrition is a common issue with Delphi studies [62].

## Conclusions

Through engaging a diverse expert stakeholder panel of physicians, nurses, pharmacists, QI experts, and patients, we were able to develop a safe opioid use agreement tailored to the surgical setting. The focus group further engaged the panel to support implementation of the agreement in the next steps of our study. The agreement cannot, and is not intended to, stand alone as the only aspect of patient education on safe opioid use after surgery and instead is intended to be used as a behavior change mechanism supplemented with additional, individualized patient education. Future research will test the effectiveness of the agreement at improving safe use, storage, and disposal of opioids prescribed to manage postoperative pain.

## Supporting information

**S1 File. Opioid agreement Delphi questionnaire.** Participants completed the Round 1 and Round 3 surveys virtually through a secure online platform.
(PDF)

**S2 File. Example feedback report to participants.** After each round, participants received an individualized feedback report that compared their responses to the summary of responses from the Delphi panel.
(PDF)

**S3 File. Education document.** A document was drafted to address the additional patient education needs identified by the participants.
(PDF)

## Acknowledgments

We would like to thank the Opioid Agreement Delphi Group for their time, effort, insight and experience: Amanda Vlcek, MHA, LCSW, Anne Bobb, Anne Stey, MD, MSc, Betsy Ross, RN, Christine Schilling, RN, Cindy Barnard, PhD, MBA, MSJS, Colleen Garrity, RN, CAPA, Daniel Kufer, David Bentrem, MD, David Kalainov, MD, MBA, Elizabeth Shepard, PharmD, BCPS, BCCCP, Gregory Auffenberg, MD, Hector Nunez, Howard Kim, MD, MS, Jeff Gardner, Jennifer Carvajal, APRN, CNP, Jill Baker, Joseph Posluszny, MD, Joshua Halpern, MD, MS, Karen Burnett, Kelly Blanchard, RN, Kevin Bajer, PharmD, BCPS, Kristen March, BS, PharmD, BCPS, Laura Batz Townsend, Laura Lane, PharmD, Linda Louis, RN, Magdy Milad, MD, MS, Michelle Starzyk, RN, Mileta Kemeza, PharmD, Morgan Aleshire, BSN, RN, CMSRN, Paloma Toledo, MD, MPH, Rachael Freeman, PharmD, BCPS, Rachel Joseph, PharmD, BCPS, Rebecca Jett, PharmD, Sterling Elliott, PharmD, BCMTMS, and Susan Jacobson. At the time of the study, all participants of the Opioid Agreement Delphi Group were affiliated with Northwestern Medicine. The group was led by Sterling Elliott, PharmD, BCMTMS (email: Sterling.Elliott@nm.org).

## Author Contributions

**Conceptualization:** Willemijn L. A. Schäfer, Julie K. Johnson, Salva N. Balbale, Jonah J. Stulberg.

**Data curation:** Willemijn L. A. Schäfer, Julie K. Johnson, Meagan S. Ager, Jonah J. Stulberg.

**Formal analysis:** Cassandra B. Iroz, Willemijn L. A. Schäfer, Julie K. Johnson, Reiping Huang, Salva N. Balbale, Jonah J. Stulberg.

**Funding acquisition:** Willemijn L. A. Schäfer, Julie K. Johnson, Jonah J. Stulberg.

**Investigation:** Willemijn L. A. Schäfer, Julie K. Johnson, Meagan S. Ager, Jonah J. Stulberg.

**Methodology:** Willemijn L. A. Schäfer, Julie K. Johnson, Reiping Huang, Jonah J. Stulberg.

**Project administration:** Cassandra B. Iroz, Willemijn L. A. Schäfer, Meagan S. Ager, Jonah J. Stulberg.

**Resources:** Jonah J. Stulberg.

**Supervision:** Julie K. Johnson, Jonah J. Stulberg.

**Writing – original draft:** Cassandra B. Iroz, Willemijn L. A. Schäfer, Julie K. Johnson, Reiping Huang, Salva N. Balbale, Jonah J. Stulberg.

**Writing – review & editing:** Meagan S. Ager.

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
