## [Decision Letter · Decision Letter 0]

9 May 2023

PONE-D-23-06761The development of a safe opioid use agreement for surgical care using a modified Delphi methodPLOS ONE

Dear Dr. Iroz,

Thank you for submitting your manuscript to PLOS ONE. After careful consideration, we feel that it has merit but does not fully meet PLOS ONE’s publication criteria as it currently stands. Therefore, we invite you to submit a revised version of the manuscript that addresses the points raised during the review process.

We look forward to receiving your revised manuscript.

Kind regards,

Ann-Sofie Sundqvist, Ph.D.

Academic Editor

PLOS ONE

4. One of the noted authors is a group or consortium [Opioid Agreement Delphi Group]. In addition to naming the author group, please list the individual authors and affiliations within this group in the acknowledgments section of your manuscript. Please also indicate clearly a lead author for this group along with a contact email address.

Reviewers' comments:

Reviewer's Responses to Questions

**Comments to the Author**

1. Is the manuscript technically sound, and do the data support the conclusions?

Reviewer #1: Yes

Reviewer #2: Yes

2. Has the statistical analysis been performed appropriately and rigorously? 

Reviewer #1: N/A

Reviewer #2: Yes

3. Have the authors made all data underlying the findings in their manuscript fully available?

Reviewer #1: Yes

Reviewer #2: Yes

4. Is the manuscript presented in an intelligible fashion and written in standard English?

Reviewer #1: Yes

Reviewer #2: Yes

5. Review Comments to the Author

Reviewer #1: The authors provide a manuscript describing the development of a safe opioid use agreement for perioperative patients using a single-center Delphi methodology.

A cursory review of literature does not demonstrate a significant volume of publications regarding safe opioid use agreements.

The severe limitation of a single-center Delphi process is acknowledged.

The manuscript is well structured but excessively wordy.

The tables and diagrams are acceptable and effectively communicate the subject matter.

The abstract does not mention thee agreed upon content w/in the agreement and should be modified in the results section to provide cursory understanding of the agreed upon content.

The methodology section is excessively long and could be trimmed to include only salient aspects of the Delphi methodology used for the study. e.g. Line 119 In Round 1, the expert panel received the survey in Qualtrics via email. This line is not necessary, and does not define Qualtrics.

Similarly, Zoom is not trademarked appropriately nor explained to the readership.

The Likert survey instrument is not defined.

Microsoft Excel is not appropriately defined nor trademarked.

A separate subheading for Data Analysis is not necessary given the lack of statistical analysis provided in this manuscript.

Line 163. The simple thematic analysis approach is not adequately explained to the readership.

Line 164. Initialing the reviewers is not necessary.

The round results should be aggressively trimmed to only include specific findings and short observations. Many of the comments should be reserved for the discussion.

Similarly, the discussion is excessively long and should be condensed. The limitations should be clearly listed without excessive defense.

Reviewer #2: Thank you for the opportunity to review the manuscript. The topic is of interest for the journal readers and an important contribution research field. Still some things needs to be added and clarified before considering publication.

General comments

Firstly, there are two general issues that needs to be considered

1. The aim of the study was to generate consensus of an opioid agreement for after surgery settings which is clear but the manuscript do not include any information about how the content of the initial agreement, that then were evaluated by the Delphi rounds, were generated. Did you generate the content yourself? Did you use another agreement as foundation? A theoretical foundation? It is strongly suggested that the authors add information concerning this issue. It is important with this information in terms of quality and reproducibility and for the reader to understand the process of the generation.

2. Concerning agreement, there were different cut of values/percentages in the different parts of the procedure, 85% and 75%. You describe that there are no strict guidelines and different recommendations for the choice of cut of which is fine, but it would be good to describe why you chose the cut of values 75% and 85%? Add information about this. Why did you use different cut of values in stage 1and 3? It seem a bit confusing to first use the cut of 85% and then, in the final stage accept lower agreement for the final version (in combination with the smaller sample). Please clarify this through all parts of the manuscript.

Introduction

You describe previous/other agreements in other settings. Did you use the content from them? Why do you need another agreement in your specific post-operation setting? What is the differences/similarities?

Aim

As the initial generation of the content of the agreement is part of the study it is suggested to add this as a part of the aim.

Methods

Page 4. Sample

The different stakeholders are clearly described but it would be good with additional information about if the participants work within public or private health care and if the participants work in the same organization/hospital, with children/adult patients or in different settings/parts of US etc. Were all stakeholders from the US or did they represent several countries/cultures? Men/women, age and for how long had they worked in this setting?

Rounds of Delphi

Page 5. As described in the general comment, more information about the generation of the initial agreement is needed.

Page 5. How was decisions made during the focus groups, consensus of by majority? Why did you choose to have one large focus group with 15 persons? This is quite a large group size for this kind of interviews and a big risk that not all participants gets the opportunity to talk, can hear all discussions, and a risk of people not feeling safe to share their thoughts. Were there discussions or just an opportunity to tell your opinion when asked? Another risk, based on the different stakeholders professions there might also be a risk of hierarchical behaviours leading to for example patients not feeling confident to discuss their experiences with for example physicians. I think this kind of risk should be addressed in the discussions limitation section.

Page 6. Se previous comment about cut of values.

Page 6. Were the interviews audio recorded? Were notes taken? If so, please add information?

Page 6. Were the design of the interviews? Open questions? Structured? Semi structured? What methodology did you follow? Is there an interview guide? Please add information about this.

Page 7, line 152. Was the Patient Education Specialist part of the research team/familiar with the agreement or unbiased? What did the results show?

Page 9. I the method you mention thematic analysis. The results is mentioned in Table 2 but it would be good to mention the results also in text. What did the themes add to you results? How do you interpret the themes?

Page 10. For round 1 you described that the participants got a reminder and that you received 100% response rate. How was this done in round 3? Add information. Also, for round 3, where the response rate were lower, which participants were not included? You write that this might have impacted the agreement ratio but it might also impact the results if for example some stakeholders were missing, this would be important to address in the limitation section.

Results

What did the analysis of the reading level show? I cannot find any information about this in the results section.

Round 1. Page 8, line 190. Here you report results on the question if they wanted to add anything. This question however s not described in the method section, pleas add all questions/items/describe the questions included in the study in the methods section.

Between round 1 and 2, it is written that minor adjustments were made based on the results. What adjustments were made? Was it minor rewording or did you replace any items?

Discussion

Page 12, line 264, I think a word is missing in the second sentence, “Our group recommends the agreement to? be used as an adjunct…”

In the background you describe previous successful agreements in similar fields. Also in page 13, line 290, you describe that some previous agreements has been less successful. I think it would be good to relate your results to previous studies. Also, if some previous agreements has not been successful, why did you want to develop agreement for your field, what’s the difference?

Strength and limitations

Page 13, line 300. Did the sample of experts represent the general population? See my comment concerning this in the method section. Could this have impact on your results?

Page 14, line 313-> It would be of importance to discuss the focus group setting. See my comment in the method section.

Page 14, line 322-323. Were there differences in age or how long the experience (years) of the stakeholders?

Page 14, line 325-327. Were the patient repetitive for the general target group/patients group?

Appendix A.

In this table you merge some of the response options together. This is not described in the method section. Add information in the method section and clarify why this was made.

Figure 1

Contains good information but is quite blurry/bad quality

Figure 2

Contains good information but is quite blurry/bad quality

6. PLOS authors have the option to publish the peer review history of their article (what does this mean?). If published, this will include your full peer review and any attached files.

Reviewer #1: No

Reviewer #2: No

---

## [Author Response · Author response to Decision Letter 0]

21 Jun 2023

Responses to Reviewer Comments (PONE-D-23-06761)

Additional Requirements

RESPONSE: We have updated the manuscript formatting to ensure it meetings the PLOS ONE style requirements including proper headings and placement of figure captions and tables. 

RESPONSE: The study was reviewed by our IRB and determined to not qualify as human subject research. Because of this, it did not require a formal consent process. However, we followed best practices in research and still informed potential participants about the purpose, procedures, risks, benefits, and voluntary nature of research. Each then individually agreed to participate. We have clarified this in the ethics section of the manuscript. 

Revision to manuscript: “Northwestern University’s Institutional Review Board reviewed the study and determined that it did not qualify as human subjects research (STU00212619). All participants were informed about the purpose of the research, procedures, potential risks and benefits, and that participation was fully voluntary and could be stopped at any time. Each expressed their consent to participate in writing for the survey and again verbally for the focus group.” (pg. 8, lines 172-176).

RESPONSE: We have made the de-identified quantitative available through our institution’s data repository (https://doi.org/10.21985/n2-12je-4d95) and the de-identified transcript of the focus group available in Qualitative Data Repository (doi:10.5064/F65NLLVI) 

4. One of the noted authors is a group or consortium [Opioid Agreement Delphi Group]. In addition to naming the author group, please list the individual authors and affiliations within this group in the acknowledgments section of your manuscript. Please also indicate clearly a lead author for this group along with a contact email address.

RESPONSE: We have added the affiliation for all authors and included the lead author for the group with their email address. 

Revision to manuscript: “At the time of the study, all participants of the Opioid Agreement Delphi Group were affiliated with Northwestern Medicine. The group was led by Sterling Elliott, PharmD, BCMTMS (email: Sterling.Elliott@nm.org).” (pg. 19, lines 350-352).

Reviewer #1

General Comment: The authors provide a manuscript describing the development of a safe opioid use agreement for perioperative patients using a single-center Delphi methodology.

RESPONSE: We thank the reviewer for their time and careful review of the manuscript. We have conducted a thorough revision of the manuscript to make the writing more concise and clarify essential aspects of the methods and results. 

1. A cursory review of literature does not demonstrate a significant volume of publications regarding safe opioid use agreements.

RESPONSE: Thank you for this comment and the opportunity to clarify the literature which supported the development of our study. There are a variety of terms in the literature including safe opioid use agreements, opioid use agreements, and opioid contracts. We have clarified this in the background section and added a few references to relevant literature on opioid agreements in various contexts. 

Revision to manuscript: “Opioid use agreements, also known as opioid contracts (27), are one tool to engage patients, are well-established in primary care (28-32), and have been shown to increase patients’ participation in their care (33) and reduce opioid use (28, 31). To date, studies on these agreements in pain clinics and primary care have shown reductions of 7% to 23% in opioid misuse (34-37), although specific effects on disposal have not been measured (38)” (pg. 3, lines 67-71).

2. The severe limitation of a single-center Delphi process is acknowledged.

RESPONSE: We have kept the limitation of the single-center study in the limitations section. 

3. The manuscript is well structured but excessively wordy.

RESPONSE: We appreciate the suggestion to make the manuscript more concise. We have made revisions throughout to cut words, improve clarity, and respond to the reviewers’ requests for elaboration on certain issues. We have removed extraneous information from the results and discussion sections, while still addressing the requests for additional clarity in certain areas. 

4. The tables and diagrams are acceptable and effectively communicate the subject matter.

RESPONSE: We have kept the tables and diagrams and uploaded new versions with higher quality per the suggestion of Reviewer 2. 

5. The abstract does not mention the agreed upon content w/in the agreement and should be modified in the results section to provide cursory understanding of the agreed upon content.

RESPONSE: We have included a brief overview of the topics covered in the agreement in our updated abstract. 

Revision to manuscript: “The final agreement included 7 items on safe use, 2 items on safe storage, and 1 item on safe disposal.” (pg. 2, lines 51-52).

6. The methodology section is excessively long and could be trimmed to include only salient aspects of the Delphi methodology used for the study. e.g. Line 119 In Round 1, the expert panel received the survey in Qualtrics via email. This line is not necessary, and does not define Qualtrics.

RESPONSE: We welcome the suggestion to make the methods section more concise and have made substantial edits to remove extraneous information while including the salient aspects of the Delphi method. For example, on page six we removed over 10 lines of text and streamlined our explanation of the rounds of the Delphi process. We also provided the appropriate reference for Qualtrics software. 

Revision to manuscript: “In Round 1, the expert panel received the survey (S1 File) via email and completed it in Qualtrics software (Qualtrics, Provo, UT).” (pg. 5, lines 123-124).

7. Similarly, Zoom is not trademarked appropriately nor explained to the readership.

RESPONSE: We have removed the references to Zoom software as we did not feel it was essential for reporting methods appropriately. 

8. The Likert survey instrument is not defined.

RESPONSE: In order to provide more clarity on the survey without adding to the length of the main text, we have added the survey as an appendix. 

Revision to manuscript: “In Round 1, the expert panel received the survey (S1 File) via email and completed it in Qualtrics software (Qualtrics, Provo, UT).” (pg. 5, lines 123-124)

9. Microsoft Excel is not appropriately defined nor trademarked.

RESPONSE: We have added the appropriate citation for Microsoft Excel software. 

Revision to manuscript: “Survey responses were exported from Qualtrics (Qualtrics, Provo, UT) to Microsoft Excel (Microsoft Corporation, Version 2304) and summarized using descriptive statistics” (pg. 7, lines 163-164).

10. A separate subheading for Data Analysis is not necessary given the lack of statistical analysis provided in this manuscript.

RESPONSE: While we did not perform advanced statistical analyses, we did conduct descriptive analyses to summarize the quantitative findings of the survey data and qualitative data analysis to summarize the key findings from the free text responses and focus group. We believe that this sub-heading is still important to orient the reader to how we analyzed our quantitative and qualitative data that we present in the results. 

11. Line 163. The simple thematic analysis approach is not adequately explained to the readership.

RESPONSE: We have added some details and clarified the citation for the simple thematic analysis and lean coding approach we took to our qualitative data analysis. 

Revision to manuscript: “We summarized the qualitative responses from the free text fields using a simple thematic analysis approach and lean coding (49, 50). Two researchers (CBI & WLS) reviewed the qualitative comments provided on the Round 1 and Round 3 surveys and the transcript from the focus group and developed a small number of codes to apply to the data. They then created a table grouping the broad codes and identified overarching themes (49). Themes and example quotes were then discussed with two additional team members (JKJ & SNB).” (pg. 7, lines 163-170).

12. Line 164. Initialing the reviewers is not necessary.

RESPONSE: We have kept the reviewers’ initials in as we believe this is an important aspect of transparency of data analysis and is recommend by qualitative study reporting guidelines such as SRQR (DOI: 10.1097/ACM.0000000000000388) and COREQ (https://doi.org/10.1093/intqhc/mzm042). 

13. The round results should be aggressively trimmed to only include specific findings and short observations. Many of the comments should be reserved for the discussion.

RESPONSE: Thank you for the suggestion to make our results section more concise. We have made substantial edits to communicate our results more efficiently. 

14. Similarly, the discussion is excessively long and should be condensed. The limitations should be clearly listed without excessive defense.

RESPONSE: Similar to the results section, we welcome the suggestion to streamline the discussion section. We have condensed the discussion section and provided more concise, clear limitations. 

Reviewer #2

General Comment: Thank you for the opportunity to review the manuscript. The topic is of interest for the journal readers and an important contribution research field. Still some things need to be added and clarified before considering publication.

RESPONSE: Thank you very much for your time and thoughtful and thorough review of our manuscript. We have addressed your points individually below and hope you agree that the revisions based on your feedback have clarified and strengthened our paper. 

Firstly, there are two general issues that needs to be considered

1. The aim of the study was to generate consensus of an opioid agreement for after surgery settings which is clear but the manuscript do not include any information about how the content of the initial agreement, that then were evaluated by the Delphi rounds, were generated. Did you generate the content yourself? Did you use another agreement as foundation? A theoretical foundation? It is strongly suggested that the authors add information concerning this issue. It is important with this information in terms of quality and reproducibility and for the reader to understand the process of the generation.

RESPONSE: Thank you for the comment and we agree that it is important to describe how the initial agreement was generated to allow for clarity and reproducibility. We have added citations to show the opioid reduction work that informed our draft agreement. 

Revision to manuscript: “First, our research team developed a draft agreement including ten items based on existing opioid use agreements used in primary care (43-47).” (pg. 5, lines 106-108).

2. Concerning agreement, there were different cut of values/percentages in the different parts of the procedure, 85% and 75%. You describe that there are no strict guidelines and different recommendations for the choice of cut of which is fine, but it would be good to describe why you chose the cut of values 75% and 85%? Add information about this. Why did you use different cut of values in stage 1and 3? It seem a bit confusing to first use the cut of 85% and then, in the final stage accept lower agreement for the final version (in combination with the smaller sample). Please clarify this through all parts of the manuscript.

RESPONSE: We thank the reviewer for this comment and recognize that our presentation of this part of the results was confusing. We defined greater than 75% to be acceptable a priori. For the focus group discussion, we decided to focus on items with less than 85% agreement because agreement was already high, and we wanted to gain feedback to further refine the lower scoring items. We have clarified this throughout the manuscript, including the abstract, methods, results, and table. 

Revision to manuscript: “In Round 1, >75% of respondents rated at least 4 out of 5 on the importance of 9 items and on the comprehensibility of 6 items.” (pg. 2, lines 47-48).

Revision to manuscript: “While greater than 75% was defined as acceptable a priori, for the focus group we chose to discuss items with less than 85% agreement because agreement was high on the Round 1 survey, and we wanted to receive feedback on how to improve the lower scoring items” (pg. 6, lines 139-142).

Revision to manuscript: “In the Round 1 survey, more than 75% of respondents rated nine items as “Important” or “Very Important” and six items as “Good” or “Very good” for comprehensibility (Table 2).” (pg. 8, lines 184-185).

Introduction

3. You describe previous/other agreements in other settings. Did you use the content from them? Why do you need another agreement in your specific post-operation setting? What is the differences/similarities?

RESPONSE: Thank you for this comment. We have added some language to the introduction to clarify why we decided to develop a new agreement for the perioperative setting, as previous agreements have primarily been used in long-term, chronic pain settings. 

Revision to manuscript: “Use of agreements in acute pain management offers a logical extension of current practices from chronic pain management. However, agreements have not been studied in the perioperative setting. Opioid use agreements have primarily been used in populations with long-term opioid use to treat chronic pain, whereas opioids prescribed following surgery are intended to treat acute short-term pain (27). Additionally, the patient-clinician relationship can differ between primary care where there are often long-term relationships versus surgery, which is characterized by more episodic, acute care. It is therefore important that we understand what is needed from patient, provider, and quality improvement (QI) perspectives in a surgery-specific context.” (pg. 3-4, lines 75-82).

Aim

4. As the initial generation of the content of the agreement is part of the study it is suggested to add this as a part of the aim.

RESPONSE: We have clarified that we aimed to develop and generate consensus on the content of a safe opioid use agreement. 

Revision to manuscript: “We therefore aimed to develop and generate consensus on the content of a safe opioid use agreement to improve safe use, storage, and disposal of opioids prescribed after surgery.” (pg. 4, lines 83-84).

Methods

Page 4. Sample. 

5. The different stakeholders are clearly described but it would be good with additional information about if the participants work within public or private health care and if the participants work in the same organization/hospital, with children/adult patients or in different settings/parts of US etc. Were all stakeholders from the US or did they represent several countries/cultures? Men/women, age and for how long had they worked in this setting?

RESPONSE: Thank you for this suggestion to clarify the setting for our expert panel. We have added details about the single health system in which we conducted the study. We did not collect demographic data from the participants. As we were interested in developing an agreement that could be used in a variety of perioperative settings, we were not aiming to be representative of any particular group. We were more interested in recruiting an expert panel based on role (e.g., surgeons, other physicians, nurses, pharmacists, QI experts, and patients/patient advocates) and specialty (i.e., orthopedics, urology, trauma, gynecology, surgical oncology, and general surgery) than by any demographic category. 

Revision to manuscript: “All clinician participants worked with adult patients at a large, private, urban academic medical center in the United States. Patient representatives were members of our health system’s Patient Family Advisory Council and were asked to participate and provide their perspective on how a patient might interpret and perceive the agreement.” (pg. 4-5, lines 101-104).

Rounds of Delphi. 

6. Page 5. As described in the general comment, more information about the generation of the initial agreement is needed.

RESPONSE: Please see response to Reviewer 2, Item #1 above for a more detailed response. We have added citations for the basis of our initial draft agreement. 

Revision to manuscript: “First, our research team developed a draft agreement including ten items based on existing opioid use agreements used in primary care (43-47.” (pg. 5, lines 106-108).

7. Page 5. How was decisions made during the focus groups, consensus of by majority? Why did you choose to have one large focus group with 15 persons? This is quite a large group size for this kind of interviews and a big risk that not all participants gets the opportunity to talk, can hear all discussions, and a risk of people not feeling safe to share their thoughts. Were there discussions or just an opportunity to tell your opinion when asked? Another risk, based on the different stakeholders professions there might also be a risk of hierarchical behaviours leading to for example patients not feeling confident to discuss their experiences with for example physicians. I think this kind of risk should be addressed in the discussions limitation section.

RESPONSE: We had multiple moderators who ensured 1) discussion of what was typed in the chat function, 2) opportunity for all participants and participant groups to talk and state their opinions, and 3) strict time management. The focus group lasted 90 minutes and with effective moderation was able to capture insights from all participants. We appreciate the possibility that the nature of relationships in medicine could theoretically impact how our participants acted in the focus group. We have added the potential bias due to hierarchical relationships to the limitations section and discussed why we used this approach. We used the polling feature of the virtual platform for statements that had no consensus on importance and the results were summarized along with the qualitative discussion then shared with all participants. 

Revision to manuscript: “In Round 2, we invited 15 members of the panel to participate in a focus group. The focus group participants were selected to ensure representation of the different stakeholder groups and surgical specialties. The group was limited to 15 participants to allow for interactive conversations. The focus group lasted 90 minutes, was conducted virtually, and audio recorded. There were four moderators who were members of the research team and ensured 1) discussion of what was typed in the chat function, 2) an opportunity for all participants and participant groups to talk and state their opinions, and 3) strict time management. The focus group was semi-structured, and the presentation slides served as the moderator guide. The moderators presented background information on surgical opioid prescribing rates and research on the importance of patient education. The participants were then shown the quantitative and qualitative responses from the Round 1 survey and asked to discuss all items where less than 85% of respondents rated at least four out of five on the Likert scale (i.e., “Good” or “Very Good” for comprehensibility and “Important” or “Very Important”). While greater than 75% was defined as acceptable a priori, for the focus group we chose to discuss items with less than 85% agreement because agreement was high on the Round 1 survey, and we wanted to receive feedback on how to improve the lower scoring items. The moderators then asked the participants open-ended questions to elicit their opinions on the importance and/or comprehensibility of each of the selected items. The moderators specifically called out participants from all stakeholder groups to ensure that all groups were represented. Participants had the opportunity to provide input through the chat function. We also used the polling function to measure agreement during the focus group, and results were summarized along with the qualitative comments and shared with participants before the Round 3 survey. The focus group concluded with a discussion on the purpose of the agreement, length, introductory text, and additional topics to cover in the agreement.” (pg. 6, lines 128-149).

Revision to manuscript: “Finally, the nature of hierarchical relationships in medicine between the various stakeholder groups might mean that some individuals did not speak freely in the focus group (e.g., patients might have resisted disagreeing with physicians). However, given the purpose of the study (i.e., consensus building) it was important to have all stakeholder groups in one focus group and we believe we mitigated the bias through the moderation process.” (pg. 18, lines 317-321).

8. Page 6. See previous comment about cut-off values.

RESPONSE: See Reviewer 2, Item #2. 

9. Page 6. Were the interviews audio recorded? Were notes taken? If so, please add information?

RESPONSE: We have clarified that the focus group was audio-recorded. Field notes were not part of our data collection. 

Revision to manuscript: “The focus group lasted 90 minutes, was conducted virtually, and audio recorded.” (pg. 6, lines 130-131).

10. Page 6. Were the design of the interviews? Open questions? Structured? Semi structured? What methodology did you follow? Is there an interview guide? Please add information about this.

RESPONSE: We have clarified that the focus group was semi-structured and included details on the various components of the focus group. 

Revision to manuscript: “The focus group was semi-structured, and the presentation slides served as the moderator guide. The moderators presented background information on surgical opioid prescribing rates and research on the importance of patient education. The participants were then shown the quantitative and qualitative responses from the Round 1 survey and asked to discuss all items where less than 85% of respondents rated at least four out of five on the Likert scale (i.e., “Good” or “Very Good” for comprehensibility and “Important” or “Very Important”).” (pg. 6, lines 134-139).

11. Page 7, line 152. Was the Patient Education Specialist part of the research team/familiar with the agreement or unbiased? What did the results show?

RESPONSE: Thank you for requesting this clarity on the review by the Patient Education Specialist. We have clarified that they were independent from the research team and added their suggestions such as using the term “medicine” instead of “medication” and including brand name as well as generic titles for Item 5. 

Revisions to manuscript: “The language of the agreement was then reviewed and edited by a Patient Education Specialist, who was independent from the research team, to reflect a sixth to eighth grade reading level.” (pg. 7, lines 152-154).

Revisions to manuscript: “The Patient Education Specialist reviewed the agreement after the focus group and made some minor wording adjustment (such as changing “medication” to “medicine) and suggested we include generic as well as brand names for medications listed in Item 5.” (pg. 14, lines 221-223).

12. Page 9. In the methods you mention thematic analysis. The results is mentioned in Table 2 but it would be good to mention the results also in text. What did the themes add to you results? How do you interpret the themes?

RESPONSE: We appreciate this suggestion to better address the qualitive data in the text of the manuscript. We have edited the results section to show the qualitative results more clearly through the three rounds. 

Revision to manuscript: “Qualitative feedback from participants revealed a desire for greater specificity, issues with comprehensibility including concerns about medical terminology such as “respiratory depression”, concerns about the effectiveness of the agreement in practice, complexity of wording, questions on the purpose of the agreement, concerns about Item 10 related the Prescription Monitoring Program (PMP), and a desire for language on shared responsibility.” (pg. 8-9, lines 186-190).

Revision to manuscript: “Themes from the focus group are summarized in Table 3. During the focus group, the panel discussed how to improve the comprehensibility for Items 2, 3, and 9. For Item 5 on opioid interactions, the group discussed the wording and layout as well as which medications were important to include in the list with potential interactions. In both Rounds 1 and 2, some participants suggested adding more detail to various items. Through discussion in the focus group, it was decided that by keeping the items more general, the agreement could be more easily adapted to different surgical specialties and practices. The participants identified additional education needs for Items 2, 3, 5, 7, and 9. Examples of such education included when, how and with whom to communicate about pain, individualized examples of potential drug interactions, and information about how to safely store and dispose of opioids. The focus group also included discussion on the potential effectiveness of the agreement, purpose of the agreement, and questions about the Prescription Monitoring Program.” (pg. 13, lines 210-220).

Revision to manuscript: “There were far fewer qualitative comments on the Round 3 than the Round 1 survey. The remaining comments contained a continued desire for specificity, comments on the improved comprehensibility with additional slight modifications, a minor formatting suggestion, and comments on the purpose of the agreement (Table 3).” (pg. 14, lines 227-231).

13. Page 10. For round 1 you described that the participants got a reminder and that you received 100% response rate. How was this done in round 3? Add information. Also, for round 3, where the response rate were lower, which participants were not included? You write that this might have impacted the agreement ratio but it might also impact the results if for example some stakeholders were missing, this would be important to address in the limitation section.

RESPONSE: Thank you for this suggestion to clarify our response rate throughout the three rounds. We have added a table that delineates the number of participants by role in each round and the response rate (Table 1). We have also clarified that the Round 3 survey was open for 1 month and participants were sent 1 reminder. 

Revision to manuscript: “The survey was open for one month and participants were sent up to one reminder.” (pg. 7, line 157).

Revision to manuscript: We added Table 1 to show the number of participants by role each round. (pg. 8, line 181).

Results

14. What did the analysis of the reading level show? I cannot find any information about this in the results section.

RESPONSE: Please see Reviewer 2, Item #11 above for the results of the Patient Education Specialist Review. 

Revisions to manuscript: “The Patient Education Specialist reviewed the agreement after the focus group and made some minor wording adjustment (such as changing “medication” to “medicine) and suggested we include generic as well as brand names for medications listed in Item 5.” (pg. 14, lines 221-223).

15. Round 1. Page 8, line 190. Here you report results on the question if they wanted to add anything. This question however is not described in the method section, please add all questions/items/describe the questions included in the study in the methods section.

RESPONSE: Thank you for this comment as we recognized we omitted details about all survey questions in the methods section. We have added details in the text as well as provided the full survey as an Appendix. 

Revision to manuscript: “We developed a survey to rate the importance and comprehensibility of each item of the draft agreement. The survey included free text fields for participants to add explanations on the importance or comprehensibility. The survey also asked if there were any topics that were not covered that they believed should be included.” (pg. 5, lines 108-111).

16. Between round 1 and 2, it is written that minor adjustments were made based on the results. What adjustments were made? Was it minor rewording or did you replace any items?

RESPONSE: Thank you for suggesting clarification on this issue. We have clarified that these edits were just in the wording. 

Revision to manuscript: “Minor updates were made in the agreement (i.e., correcting spelling and grammar) prior to the Round 2 focus group.” (pg. 9, lines 191-192).

Discussion

17. Page 12, line 264, I think a word is missing in the second sentence, “Our group recommends the agreement to? be used as an adjunct…”

RESPONSE: In condensing the results and discussions sections we removed this line. We appreciate the reviewer’s attention to detail and helping us clarify the language throughout. 

18. In the background you describe previous successful agreements in similar fields. Also in page 13, line 290, you describe that some previous agreements has been less successful. I think it would be good to relate your results to previous studies. Also, if some previous agreements has not been successful, why did you want to develop agreement for your field, what’s the difference?

RESPONSE: We thank the reviewer for this comment and agree that we could provide more information on how our work built off previous studies on opioid use agreements and how our agreement for the surgical context was new and important. We have added information to the discussion section. 

Revision to manuscript: “Other studies have found limited effectiveness of opioid agreements in primary and chronic care settings (29, 38, 56). Limitations of these previous studies include inconsistent use of the agreement, sampling bias, lack of a consistent definition of opioid misuse or abuse, and few studies provided sufficient description or shared the actual text of their agreement. Approaches to reducing opioid addiction have been categorized as primary (preventing new cases of opioid addiction), secondary (identifying early cases of opioid addiction), and tertiary (ensuring access to effective addiction treatment) (57). Previous studies have mostly focused on secondary prevention, working with populations who chronically use opioids, whereas this study was intended to develop an agreement for primary prevention, reducing the risks for patients prescribed opioids for acute pain management. To increase the likelihood our agreement would be effective, we followed a robust Delphi process to develop content that was relevant to the patient population and easy to understand. We engaged a diverse panel of key stakeholders from various professions and specialties to generate consensus on the content of the agreement and had the language reviewed by a Patient Education Specialist. The effectiveness of our agreement is still unknown. The next step for our research team is to study the effectiveness of the agreement at improving safe use, storage, and disposal of opioids prescribed for postoperative pain management.” (pg. 17, lines 288-303).

Strength and limitations

19. Page 13, line 300. Did the sample of experts represent the general population? See my comment concerning this in the method section. Could this have impact on your results?

RESPONSE: Thank you for this comment about generalizability of the results. Our expert panel included surgeons, nurses, pharmacists, QI experts, other physicians, patients, and patient advocates from a variety of surgical specialties including orthopedics, urology, trauma, gynecology, surgical oncology, and general surgery. The target population for the safe opioid use agreement was patients undergoing any kind of surgery who were prescribed opioids to treat their postoperative pain. The agreement was developed to be used broadly through various surgical fields, so the concept of a general population is difficult to define in this scenario. The agreement could be used in variety of populations that vary in clinical and demographic characteristics. Likewise, the people who treat surgical patients could vary greatly in their personal demographics, practice settings, and years of experience. For those reasons, it is not possible to state if the panel is representative or not. It is possible, and perhaps likely, that a different group of experts with the same roles would have come to a slightly different consensus on the exact content of a safe opioid use agreement for surgery. We have addressed this in the limitations section. However, a strength of our study is that we had a broad range of experts from a variety of fields, not just opioid prescribers, and we included patients. This broad approach makes it more likely that the agreement could be used, and perhaps adapted, in a variety of surgical settings. Future studies will test the effectiveness of the agreement in various settings with different patient populations. 

Revision to manuscript: “Second, we only sought experts from one healthcare system, which perhaps limits the applicability in other settings. This focus on our own healthcare system was purposeful, as we wanted to develop, implement, and test the agreement locally before disseminating to other settings. Reliability and validity of results from Delphi studies have been questioned, but including experts in the field of study strengthens the validity (41, 58, 59). It is possible, and perhaps likely, that a similar experts panel with different participants would have developed a slightly different final agreement.” (pg. 17-18, lines 306-312). 

Revision to manuscript: “Future research will test the effectiveness of the agreement at improving safe use, storage, and disposal of opioids prescribed to manage postoperative pain.” (pg. 19, lines 337-338). 

20. Page 14, line 313-> It would be of importance to discuss the focus group setting. See my comment in the method section.

RESPONSE: Please see Reviewer 2, Item #13 above. We included a sub-sample of the expert panel for the focus group so there would be opportunity for discussion. We have added Table 1 to show how many people from each stakeholder group participated in the focus group. (pg. 8, line 181).

21. Page 14, line 322-323. Were there differences in age or how long the experience (years) of the stakeholders?

RESPONSE: We did not collect age or years of experience demographics from the participants. As described in more detail in the comment below (Reviewer 2, Comment 22), we recruited a broad range of clinicians and patients with the goal of developing an agreement that could be applied to a variety of surgical populations, without the goal of being representative of a particular population. 

22. Page 14, line 325-327. Were the patient repetitive for the general target group/patients group?

RESPONSE: The target patient population for the safe opioid agreement was adults undergoing surgery. We purposefully kept this population broad so that the agreement could be applicable to a variety of perioperative settings. We engaged clinicians from a variety of surgical specialties, including orthopedics, urology, trauma, gynecology, surgical oncology, and general surgery with the goal of developing an agreement that was broadly applicable to acute, post-surgical pain management. The patient representatives were engaged in our Patient Family Advisory Council and were interested in working on a project related to opioid reduction in surgery. We did not specifically ask about their medical conditions or experiences with surgery but instead asked them to give us their perceptions on how they thought patients might interpret and perceive the agreement. 

Revision to manuscript: “Patient representatives were members of our health system’s Patient Family Advisory Council and were asked to participate and provide their perspective on how a patient might interpret and perceive the agreement.” (pg. 4-5, lines 102-104).

Appendix A.

23. In this table you merge some of the response options together. This is not described in the method section. Add information in the method section and clarify why this was made.

RESPONSE: We have clarified in the abstract, methods, and results that the results we provide are the percentage of respondents who rated the item at least 4 out of 5 on the Likert scale (Important or Very Important and Good or Very Good). 

Revision to manuscript: “The participants were then shown the quantitative and qualitative responses from the Round 1 survey and asked to discuss all items where less than 85% of respondents rated at least four out of five on the Likert scale (i.e., “Good” or “Very Good” for comprehensibility and “Important” or “Very Important”).” (pg. 6, lines 136-139).

Revision to manuscript: “In the final agreement, we included items for which more than 75% of participants rated at least four out of five on the Likert scale for importance and comprehensibility.” (pg. 7, lines 157-159).

Revision to manuscript: “In the Round 1 survey, more than 75% of respondents rated nine items as “Important” or “Very Important” and six items as “Good” or “Very good” for comprehensibility (Table 2).” (pg. 8, lines 184-185).

Revision to manuscript: “All ten items met the final threshold for inclusion with at least 75% of respondents reporting the item as “Important” or “Very important” and “Good” or “Very good” for comprehensibility (Table 2).” (pg. 14, lines 225-227).

Figure 1

24. Contains good information but is quite blurry/bad quality

RESPONSE: Thank you for pointing this out. We have included a higher resolution image in our resubmission. 

Figure 2

25. Contains good information but is quite blurry/bad quality

RESPONSE: Thank you for pointing this out. We have included a higher resolution image in our resubmission.

---

## [Decision Letter · Decision Letter 1]

15 Aug 2023

PONE-D-23-06761R1The development of a safe opioid use agreement for surgical care using a modified Delphi methodPLOS ONE

Dear Dr. Iroz,

Thank you for submitting your manuscript to PLOS ONE. After careful consideration, we feel that it has merit but does not fully meet PLOS ONE’s publication criteria as it currently stands. Therefore, we invite you to submit a revised version of the manuscript that addresses the points raised during the review process.

I agree with both reviewers regarding that you manuscript has improved after your first revision. In order meet PLOS ONEs requrements and to get your manuscript accepted you nevertheless need to meet all of the requirements that reviewer no 2 has pointed out. 

We look forward to receiving your revised manuscript.

Kind regards,

Ann-Sofie Sundqvist, Ph.D.

Academic Editor

PLOS ONE

Journal Requirements:

Reviewers' comments:

Reviewer's Responses to Questions

**Comments to the Author**

1. If the authors have adequately addressed your comments raised in a previous round of review and you feel that this manuscript is now acceptable for publication, you may indicate that here to bypass the “Comments to the Author” section, enter your conflict of interest statement in the “Confidential to Editor” section, and submit your "Accept" recommendation.

Reviewer #1: All comments have been addressed

Reviewer #2: (No Response)

2. Is the manuscript technically sound, and do the data support the conclusions?

Reviewer #1: Partly

Reviewer #2: Yes

3. Has the statistical analysis been performed appropriately and rigorously? 

Reviewer #1: Yes

Reviewer #2: Yes

4. Have the authors made all data underlying the findings in their manuscript fully available?

Reviewer #1: Yes

Reviewer #2: Yes

5. Is the manuscript presented in an intelligible fashion and written in standard English?

Reviewer #1: Yes

Reviewer #2: Yes

6. Review Comments to the Author

Reviewer #1: none at this time

Reviewer #2: The manuscript has improved significantly and most comments from the previous peer-review have been carefully handled. However, there are still some issues in the method section that need to be improved.

Method

Page 5-7, Rounds of Delphi study, I recommend that you use headlines in this section as it is a bit hard to follow the methods in the current format. Suggestions of headlines are for example Preparations (where you describe how the first version of the agreement were developed), Pilot testing, Delphi Round 1, Delphi round 2, and Delphi round 3.

Page 5, line 106-111. The addition concerning that the authors developed the draft agreement is good but to understand the process it would be good with more info concerning why these agreements were used, why you choose the specific items and how you reached agreement in the group.

Page 5, section 2. Here you write about a pilot study. As this isn’t part of the Delphi round this section is a bit unclear. Why did you do this pilot testing before the Delphi rounds? Is it suggested in the literature?

Page 7, data analysis. This section is a bit confusing, you write that you used descriptive statistics and then summarized the qualitative responses from the free text fields. So were the thematic analysis made just from Delphi round 1 and 3 where the participant answered the survey in free text? If not this must be clarified. If yes, how did you analyze round 2, in the qualitative interviews? What did you do with the audio recordings from the interviews? Were they transcribed and analyzed? This must be clarified both for the reader and for the replicability.

Page 7. How was the thematic analyses made, add info about the steps.

Table 3. I like the addition of Table 3 as it contributes with info about the results of the thematic analysis. It would be good with more info about the analysis step and it could be helpful with an example of how the analysis was made from full text to a theme. For example there seems to be some overlap between the themes, for example formatting and connotation of certain words could perhaps be part of comprehensibility? How did you argue when you did the analyses and how do you handle the overlaps.

7. PLOS authors have the option to publish the peer review history of their article (what does this mean?). If published, this will include your full peer review and any attached files.

Reviewer #1: No

Reviewer #2: No

---

## [Author Response · Author response to Decision Letter 1]

6 Sep 2023

Responses to Reviewer Comments (PONE-D-23-06761R1)

Reviewer #2

General Comment: The manuscript has improved significantly and most comments from the previous peer-review have been carefully handled. However, there are still some issues in the method section that need to be improved.

RESPONSE: Thank you very much for your time and thorough review of our revised manuscript. We appreciate the additional feedback and have addressed each point below. 

1. Page 5-7, Rounds of Delphi study, I recommend that you use headlines in this section as it is a bit hard to follow the methods in the current format. Suggestions of headlines are for example Preparations (where you describe how the first version of the agreement were developed), Pilot testing, Delphi Round 1, Delphi round 2, and Delphi round 3.

RESPONSE: Thank you for this suggestion. We have added sub-headings in the methods section to make it easier to follow. 

Revision to manuscript: “Pilot testing” (pg. 5, line 120); “Round 1” (pg. 6, line 129); “Round 2” (pg. 6, line 135); “Round 3” (pg. 7, line 163)

2. Page 5, line 106-111. The addition concerning that the authors developed the draft agreement is good but to understand the process it would be good with more info concerning why these agreements were used, why you choose the specific items and how you reached agreement in the group.

RESPONSE: We have added some more information about the process of developing the draft agreement for the Round 1 survey. The sample opioid agreements cited in the text were chosen based on what was publicly available and their applicability to the surgical setting. Prior surgical opioid reduction initiatives within our healthcare system guided our research team in understanding what was needed in an opioid use agreement for surgery and choosing items (e.g., including disposal of leftover opioids). Our research team discussed the publicly available agreements, items important for the surgical setting, and developed the content for draft agreement through group discussions. This then served as a starting point for the Delphi study where participants were asked if the items were important to include and if there were other topics, not in the original draft agreement, that should be included as well. 

Revision to manuscript: “Items included in the draft agreement were chosen through discussions within the research team and were based on publicly available opioid agreements and our previous research on surgical opioid reduction within our health system. For example, our previous work showed that communicating and setting expectations about pain relief as well as discussing safe disposal of leftover opioids were important areas for improvement, so they were included in the draft agreement (48, 49).” (pg. 5, lines 108-112).

3. Page 5, section 2. Here you write about a pilot study. As this isn’t part of the Delphi round this section is a bit unclear. Why did you do this pilot testing before the Delphi rounds? Is it suggested in the literature?

RESPONSE: Thank you for this comment. As we had previously mentioned in the discussion section (pg. 19, line 342), pilot testing the survey is a recommended part of a Delphi study. We have clarified this in the methods section and added the citations to support conducting a pilot study before the first round of the Delphi study. 

Revision to manuscript: “Pilot testing is recommended before starting a Delphi study (41, 48).” (pg. 5, lines 121).

4. Page 7, data analysis. This section is a bit confusing, you write that you used descriptive statistics and then summarized the qualitative responses from the free text fields. So were the thematic analysis made just from Delphi round 1 and 3 where the participant answered the survey in free text? If not this must be clarified. If yes, how did you analyze round 2, in the qualitative interviews? What did you do with the audio recordings from the interviews? Were they transcribed and analyzed? This must be clarified both for the reader and for the replicability.

RESPONSE: We have clarified that we recorded and transcribed the Round 2 focus group. We conducted the same process of thematic analysis for the qualitative data from all three rounds (free text for Rounds 1 and 3 and transcribed from the focus group in round 2) and have clarified this in the text. 

Revision to manuscript: “The focus group lasted 90 minutes, was conducted virtually, and was audio recorded and transcribed for analysis.” (pg. 6, lines 138-139).

Revision to manuscript: “Survey responses were exported from Qualtrics (Qualtrics, Provo, UT) to Microsoft Excel (Microsoft Corporation, Version 2304) and quantitative values were summarized using descriptive statistics. Qualitative responses from the free text fields in the Round 1 and Round 3 surveys as well as the transcript of the focus group from Round 2 were summarized using a simple thematic analysis approach and lean coding (52, 53). First, two researchers (CBI & WLS) reviewed all qualitative comments provided on the Round 1 and Round 3 surveys and the transcript from the focus group. Second, the two researchers discussed the qualitative data to develop codes inductively which were applied to the data through group discussion. For example, quotes relating to desired reading level of the items were labeled as “reading level”. Third, they created a table grouping the broad codes to identify overarching themes (52). For example, the “reading level” code was assigned to the theme “comprehensibility”. Finally, themes and example quotes were discussed with two additional team members and refined for clarity (JKJ & SNB).” (pg. 7-8, lines 172-183).

5. Page 7. How was the thematic analyses made, add info about the steps.

RESPONSE: Please see comment 6 below. 

6. Table 3. I like the addition of Table 3 as it contributes with info about the results of the thematic analysis. It would be good with more info about the analysis step and it could be helpful with an example of how the analysis was made from full text to a theme. For example there seems to be some overlap between the themes, for example formatting and connotation of certain words could perhaps be part of comprehensibility? How did you argue when you did the analyses and how do you handle the overlaps. 

RESPONSE: We have provided a more detailed description of the four-step process we conducted, including reviewing the data, developing codes, categorizing codes into themes, and receiving feedback on the final themes. 

Revision to manuscript: “Qualitative responses from the free text fields in the Round 1 and Round 3 surveys as well as the transcript of the focus group from Round 2 were summarized using a simple thematic analysis approach and lean coding (52, 53). First, two researchers (CBI & WLS) reviewed all qualitative comments provided on the Round 1 and Round 3 surveys and the transcript from the focus group. Second, the two researchers discussed the qualitative data to develop codes inductively which were applied to the data through group discussion. For example, quotes relating to desired reading level of the items were labeled as “reading level”. Third, they created a table grouping the broad codes to identify overarching themes (52). For example, the “reading level” code was assigned to the theme “comprehensibility”. Finally, themes and example quotes were discussed with two additional team members and refined for clarity (JKJ & SNB).” (pg. 7-8, lines 174-183).

---

## [Editor Report · Decision Letter 2]

10 Sep 2023

The development of a safe opioid use agreement for surgical care using a modified Delphi method

PONE-D-23-06761R2

Dear Dr. Iroz,

We’re pleased to inform you that your manuscript has been judged scientifically suitable for publication and will be formally accepted for publication once it meets all outstanding technical requirements.

Kind regards,

Ann-Sofie Sundqvist, Ph.D.

Academic Editor

PLOS ONE

Additional Editor Comments (optional):

Dear Dr Cassandra Bryce Iroz,

I would like to draw your attention to that the colour scheme used in Table 2 is difficult to understand (i.e. the red and green). I recommend that you describe what the colours stand for either in the main text and as a footnote in the table, or only as a footnote in the table.

Kind regards

Ann-Sofie Sundqvist
---

## [Editor Report · Acceptance letter]

18 Sep 2023

PONE-D-23-06761R2 

The development of a safe opioid use agreement for surgical care using a modified Delphi method 

Dear Dr. Iroz:

I'm pleased to inform you that your manuscript has been deemed suitable for publication in PLOS ONE. Congratulations! Your manuscript is now with our production department. 

Kind regards, 

on behalf of

Dr. Ann-Sofie Sundqvist 

Academic Editor

PLOS ONE